# Intergenerational Deliberations for Long Term Sustainability

**Llinos Haf Spencer** [1,*] **, Mary Lynch** [2] **, Gwenlli Mair Thomas** [1] **and Rhiannon Tudor Edwards** [1]

1   Centre for Health Economics and Medicines Evaluation, Bangor University, Gwynedd LL57 2PZ, UK
2   School of Health and Life Sciences, University of West Scotland, Glasgow G72 0LH, UK
*   Correspondence: l.spencer@bangor.ac.uk

**Abstract:** Grŵp Cynefin, a social housing association in North Wales, United Kingdom (UK) with other partner organisations, had a vision to create a community Hub in the Nantlle Valley to strengthen and support the health and well-being of the local community through the provision of a range of traditional and preventative services. Social prescribing (SP), which is a non-medical support using community assets, would be a part of this new innovative Hub. SP activities would be co-designed and co-produced by current community members. Drawing on the principles of citizens' assembly deliberations and Future Design, four focus groups (n = 16) were conducted to develop sustainable strategies for SP activities as part of the proposed Hub. Deliberations on the perspectives of future generations were considered along with current community needs. Findings from the focus groups imply that current members of society are open to the concept of taking an inter-generational approach when designing SP activities to address the social and economic needs of the community along with integration of traditional and preventative community health services. Deliberations highlighted that the proposed Hub could strengthen communities and support community health and well-being, by providing a place to socialise and acting as a single point of access for community services, which could promote social cohesion in line with the Well-being for Future Generations (Wales) Act. Applying a long-term thinking approach to citizens' assembly deliberation design offers a voice to the interests of future generations, providing inter-generational equity.

**Keywords:** social prescribing; co-production; co-design; well-being; health equity; social determinants of health; healthy people programs

## 1. Introduction

Social prescribing (SP) is about connecting citizens to community support to promote self-management of their health and well-being with individuals signposted to non-medical interventions/activities [1]. Social prescribing focuses on what is important to the individual using community assets to empower the individual to promote self-management and improve health and well-being outcomes [2]. SP referrals come from a range of sources such as GP surgeries, pharmacies, voluntary and community sectors, and self-referrals. SP interventions/activities can therefore offer patients the time and resources that health professionals do not have, to overcome their challenging situations and concerns and co-design their personalised social prescription. SP interventions/activities also accord with the Welsh Government's emphasis within 'A Healthier Wales strategy' [3] on the importance of connecting people with community activities to enable them to remain active, reduce loneliness and isolation, and support mental and physical health. In addition, the Welsh Government's 'Programme for Government' places an emphasis on introducing a national framework for SP to tackle loneliness as a component of their aim to "provide effective, high-quality and sustainable healthcare" [4] (p. 3).

In the Nantlle Valley, North Wales, there is an ambitious Hwb project led by Grŵp Cynefin (a housing association providing affordable social homes). The vision for the Hub is that it would be an innovative community hub to serve the village and the communities of the Nantlle Valley and beyond. The Hub's aim was to strengthen communities across

the valley, supporting people's health and well-being through the provision of a range of tradition and preventative services. In addition, the Hub would offer a place to socialise and connect people and be a single point of access to health, housing, and community and council services, which aims to strengthen the local economy. These key elements inextricably link with the Well-Being of Future Generations (Wales) Act 2015 [5] which legislates for a prosperous Wales, a resilient Wales, a healthier Wales, a more equal Wales, a Wales of cohesive communities, a Wales of vibrant culture and thriving Welsh language, and a globally responsible Wales. This Act places a well-being duty on all public bodies to adopt the principle of sustainable development as they aim to achieve seven well-being goals for future generations, including building a healthier Wales, a more equal Wales, and a Wales of cohesive communities [6]. The potential for the Hub is that it would include integrated services, including core general medical facilities including a GP surgery, a dental service, and a pharmacy. In addition, the Hub would accommodate community health services, a 36-bed residential home, and 17 flats that accommodate independent living. Multipurpose spaces for young people would also be part of the Hub as well as support services and spaces for third sector/voluntary organisations to deliver outreach provision. Overall, the Hub would be a space to promote intergenerational activity. In addition, the aspiration would be that the whole site, would include green spaces, with the aim to be net zero using locally sourced sustainable materials with the circular economy principles in mind.

To determine if a sustainability approach to the development of the proposed Hub is viable, conversations with community members were considered imperative to identify current needs whilst taking account of long-term sustainability for future generations [6]. Conversations with Grŵp Cynefin established that working towards holistic health and well-being provision is part of the Hub's vision.

The Future Generations Commissioner for Wales (2020) [7] emphasizes involving citizens in sustainable developments using two-way conversation methods that inspire and enthuse. An investigation into data collection methods that inspire long-term, sustainable ideas among participants revealed the Future Design citizen assembly's movement in Japan. Citizen assemblies are a form of deliberative democracy in which a representative sample of a population gathers to learn about and discuss policy issues, leading to policy recommendations [8,9]. Citizen assemblies usually reconvene on more than one occasion, meaning that deliberations occur over time, giving participants sufficient time to learn and reflect on long-term issues [10]. The significance of this approach to research is that it integrates a citizens' assembly design to participatory deliberations in 'real-world' research. This approach encourages long-term thinking among future generation focus group participants and takes account of current needs whist also integrating the needs of future generations for long-term sustainability of resources.

Future Design is a recent framework developed by the economist Ttsuyoshi Saijo. The ethos is that we live in a world where human activity creates "future failures" such as global warming, loss of biodiversity, and outstanding debt in many countries [11] (p. 1). Future Design aims to activate a human trait called "futurability", where people feel the happiness of having acted in a way that benefits future generations [12] (p. 8). Research demonstrates that the application of intergenerational accountability (IA) in research design can facilitate empowering individuals to select sustainable alternatives. When considering sustainability issues IA applied as a social tactic can generate a mechanism which provides a depiction of reasons which shaped and influenced preceding generational choices as well as justifications for previous choices, while validating succeeding generations to be provisional allies by means of perceived motivations and intelligence gleaned [13]. Evidence indicates that the concept of intergenerational sustainability dilemma (ISD) can be effectively applied to key significant challenges such as climate change. The application of ISD can influence the public to reflect on current obligations, fairness, and equality while considering sustainable options for future generations. However, it is acknowledged that ISD can be influenced by private enterprise and democracy while not considering the needs

of future generations [14]. It is established that deliberation can shape and influence change in an individual's opinion, and the use of ISD extends short-term thinking in considering sustainable options rather than imposing a cost on future generations. It is acknowledged that the impact of ISD is shaped by an individual's inherent base opinion. However, intragenerational deliberation does not automatically influence individual opinions in solving sustainability issues for the future [15].

Conversely, this evidence is at odds with findings which indicate that taking the point of view of future generations could have an impact on an individuals' thinking and preferences, future benefits and potential economic incentives for sustainable initiatives [16]. Cathedral thinking [10], future design, and futurability are innovative theories for long-term thinking and sustainability [17]. Relinquishing current benefits and utilities to enhance future generation gains through forward-thinking decisions should be integrated into research design to capture sustainability, fostering long-term thinking to explicitly account for ecological preferences for future generations.

There are several different mechanisms within Future Design to try to encourage this trait [18], and this current study will draw upon the principles of the future-ahead-and-back (FAB) mechanism. The FAB mechanism requires participants to imagine the perspectives and emotions of future generations. Participants are then invited to return to consider the present time, with the hope that considering future generations beforehand will lead them to discuss and decide upon the most sustainable actions [8,19]. It is acknowledged that applying intergenerational sustainability dilemma (ISD) to FAB can influence decisions regarding considering future generations and the potential consequences of the decisions. Evidence suggests that applying an ISD approach along with the FAB paradigm built into Future Design can promote implicit trade-offs when making decisions, and can facilitate reciprocity [8,18]. Evidence of the use of FAB under laboratory conditions shows that considering future generation perspectives before making a decision leads individuals to make the most sustainable decision [20], including pro-self-individuals from a capitalist society [21].

The aim of this study was to examine the opportunities and barriers to developing SP interventions/activities which consider future generations when establishing wellness and well-being outcomes. The objectives of this investigation were to (1) determine among the community of the Nantlle Valley if the development of the SP interventions/activities could improve health and well-being outcomes, (2) to examine barriers and opportunities for SP intervention development, and (3) to understand if the development of a new health and well-being Hub had the potential to improve health and well-being outcomes among the community.

## 2. Materials and Methods

### 2.1. Design

Data from the Nantlle Valley community was collected using four focus groups. The focus group method was chosen for data collection for the current study to bring a group of individuals living in the Nantlle Valley together to answer a set of questions and discuss. A convenience sampling method was applied as it is considered an appropriate method for recruiting a range of participants across various age groups, in an effort to gain a naturalistic, holistic view of the multiple realities in the Nantlle Valley [22,23]. Due to COVID-19, the purposeful sampling method relied on online platforms due to COVID-19 social distancing restrictions. Therefore, the sampling process relied on individuals taking notice of emails and social media advertisements, hence the sampling method required applying a convenience approach.

Participatory deliberative methods, comprising citizens' assemblies, are obtaining enhanced attention worldwide [24]. The citizens' assembly approach is advocated as a means towards long-term thinking for key issues to discuss projects beyond a human life time [10]. Applying a long-term thinking approach taking account of future generations, has been incorporated into democratic processes in deliberation assemblies used to shape

policies. These citizens' assembly members are guardians of the future, safeguarding the interests of future generations. Applying long-term thinking accounts for future generations and ensures inter-generational equity [10].

*2.2. Participants*

Recruitment to focus groups was by means of emails, social media, advertisement via online newsletter, and local papers. A total of 16 participants consented to contribute to the focus groups Today (n = 7) and Future generations (n = 9). Due to the COVID-19 pandemic focus group meetings were held via the online platform Zoom. Drawing on the principles of citizens' assembly deliberations and Future Design, four focus groups (n = 16) were conducted to develop sustainable strategies for SP activities as part of the proposed Hub. The citizens' assembly approach was applied to bring together a group of people to learn and discuss the key issues facing the community of the Nantlle Valley and to understand what they think should happen. The focus groups were made up of members who were representative of the wider population of the Nantlle Valley. In two of the focus group groups, the language of communication was Welsh, and in the other two, the language of communication was English, reflecting the linguistic nature of the area under investigation. Participants were aged 21–80 years old, most participants (n = 12) identified as females and the remaining (n = 4) identified as male. Nobody identified as 'other' in terms of gender.

Participants were recruited over a period of one month (18 January 2021–18 February 2021). The aim of this approach was to determine if a long-term thinking approach to conversations with community members could identify long-term well-being needs and sustainability. Conversations with Grŵp Cynefin established that working towards holistic health and well-being provision is part of the Hub's vision, to work towards the sustainable development of a healthy, resilient community. See Table 1.

**Table 1.** Number of participants within each focus group.

| Type of Focus Groups | Number |
|---|---|
| Number of participants Today Group (Welsh medium) | 5 |
| Today Group (English medium) | 2 |
| Future generations Group (Welsh medium) | 7 |
| Future generations Group (English medium) | 2 |
| Total | 16 |

*2.3. Materials*

A novel approach was applied to the Nantlle Valley focus groups schedules to generate data that would provide insight into the long-term needs and requirements of the communities using the planned health and well-being Hub in Penygroes, Nantlle Valley. This included developing two focus groups schedules presented and explained in Tables 2–4. One focus group schedule was titled the 'Today Group' and facilitated a deliberation on the present generation (short-termism also known as short term thinking) [10]. The second focus group schedule was titled the 'Future generations Group' was aimed to activate long-term thinking among participants and the deliberation on future generations' perspective (forward and back thinking mechanism (FAB)) [18].

The questions for both focus group schedules were based on the "Good Ancestor Conversations" principles developed by Roman Krznaric (2020) (p. 242). Recognizing that our actions today affect the quality of life of future generations, Kzarnic (2020) encourages collective long-term thinking and planning (citizens' assembly deliberation). The philosophy of these principles shapes good ancestor conversations to facilitate long-term thinking and generate ideas on how to lead sustainable lives for the benefit of future generations. The focus group schedule for the 'Today Group' is outlined in Table 3 and the 'Future generations Group' focus group schedule is outlined in Table 4. Both Today and Future

generations focus groups were recorded on Zoom and moderated by the researcher (GT). Community representatives associated with Grŵp Cynefin helped to facilitate the group.

**Table 2.** Focus group titles and their meanings.

| | |
|---|---|
| **Today Group(s)** | Participants were asked to answer questions that focused on the present generations' perspective—the health and well-being issues and needs affecting Nantlle Valley residents today. |
| **Future generations group(s)** | Drawing upon the future-ahead-and-back mechanism [21] participants were asked questions that prompt long-term thinking (the Nantlle Valley in 100–200 years) and produce responses which will make explicit the mechanisms required today to design a robust and resilient SP intervention that will lead to sustainable well-being outcomes for future generations |

**Table 3.** Today group schedule.

| | |
|---|---|
| Introductions: | What Is Your Name and in Which Village within the Nantlle Valley Do You Currently Live in? |
| Opening questions: | 1. What does well-being mean for you? <br> 2. What is your understanding of social prescribing? <br> 3. What do you know about the well-being Hub under development in the Nantlle Valley? |
| Key questions: | 4. Are you aware of any current social prescribing or well-being services available in the Nantlle Valley? |
| | 5. Do you think that these services have been welcomed among the community? |
| | 6. Are you aware of any opportunities in the Nantlle Valley to develop new social prescribing well-being services/groups/interventions, e.g., developing allotments/men sheds on unused green spaces? |
| | 7. A key aim of the new health and well-being hub is that GP's will be able to refer patients to SP interventions within the community. Would you take part in SP interventions if offered? |
| | 8. What do you think would be a challenge for you to participate in an SP intervention in the community? |
| | 9. What do you think will be the long-term impact of the COVID 19 pandemic on the community and delivery of health and well-being interventions? |
| | 10. What do you think Grŵp Cynefin could put in place/include now when developing the new health and well-being hub to improve the service? |
| Ending question: | 11. Thinking about the needs of the Nantlle Valley community now what suggestions would you think should consider in the development of the new health and well-being hub? |

Ethical approval for this study was granted by Bangor University's Healthcare and Medical Sciences Academic Ethics Committee (2020–16850) on 11 January 2021. Due to the COVID-19 pandemic restrictions, the participants were sent an electronic consent form to return before the focus group and their verbal consent were recorded at the beginning of each focus group.

**Table 4.** Future generations group schedule.

| Introductions | Question | Good Ancestor Principles |
|---|---|---|
| Opening questions: | 1. What does well-being mean for you?<br>2. What do you know about the well-being Hub under development in the Nantlle Valley? | # |
| | Short term thinking is about dealing with health and well-being services now and not about sustainability for the future. Long term thinking is realizing that we are a dot on the timeline, and we need to be thinking towards the end of the line. | |
| | 3. What for you are the most powerful reasons for caring about the future generations who will be living in the Nantlle Valley beyond your lifetime? | Intergenerational justice |
| | 4. What kind of community do want future generations to inherit from the present generation? | Future generations Mindset |
| | 5. What is worth fighting for to secure the future generation's health and well-being? | Deep time humility |
| | 6. How can we sustain the resources of the Nantlle Valley and ensure that they are passed on to future generations that will live in the Nantlle Valley?<br>(Resources can refer to natural resources, services, the community, etc.) | |
| Key questions: | 7. What long term projects could you pursue with others that could extend beyond your own lifetime to secure the well-being of future generations? | Cathedral Thinking |
| | 8. Think about the future. Do you anticipate a different pathway for holistic health and well-being interventions or services in the Nantlle Valley? Holistic health and well-being services take full account of the person's situation, not just treat symptoms, e.g., increased IT interventions (increased use of technology) or different lifestyle choices such as health and well-being projects. | Holistic Forecasting |
| | 9. What do you think should be the ultimate goal of the health and well-being Hub in the Nantlle Valley for future generations? | Transcendent goal |
| Ending question: | 10. When thinking about the needs of future generations in the Nantlle Valley is there anything that we have not already discussed that is important for Grŵp Cynefin to consider and include in the development of the new health and well-being Hub? | |

*2.4. Analytic Method*

The study team consisted of bilingual Welsh and English speakers (three of the four authors are fluent in Welsh) and were therefore able to provide an active offer of Welsh to the Welsh speakers involved in the study [25]. All recorded focus groups were transcribed in the original language (Welsh or English). The transcripts were then coded using a bilingual coding frame (Welsh/English) and analysed according to identified a priori themes [26], which were adapted based on the qualitative data collected. Three authors were involved

in the data analysis process (GT, ML, and LHS) and agreed on the theoretical framework findings presented in the results section.

## 3. Results

This focus group study set out to determine the efficiency of current SP interventions taking place in the Nantlle Valley and to identify the specific local community needs requirements for the future and long-term sustainability. The purpose of the focus groups was to gather information about co-production and perceptions regarding the need for co-produced SP interventions in the Nantlle Valley. Focus group questions were developed by the bilingual research team to investigate if the new health and well-being Hub had the potential to improve the health and well-being opportunities in the community. The Today and Future generations qualitative focus group data were analysed using a thematic framework approach [26] to analyse the data. Four themes were identified: Current SP interventions in the Nantlle Valley; opportunities for new social prescribing interventions; possible barriers to the development of co-designed and co-produced social prescribing interventions; and community needs and strategies.

### 3.1. Current SP Interventions in the Nantlle Valley

In terms of current SP interventions taking place in the Nantlle Valley the SP consists of a link worker who is taking referrals from the GP surgery and other surgeries in the area. Focus groups reference was made to various community exercise and leisure groups that are taking place in various community venues, for different age groups. The results suggest that such groups and activities do contribute positively to individuals' well-being and develop reciprocal relationships between service providers and users which can be a transformative process [27]. This is due to how participants during all focus groups seemed grateful and proud of such provision and expressed their regret that some participants were not aware of some or all the SP resources available.

> "There are things going on in the Valley and people aren't aware of them [...]
> I would not have known about [an event] if it wasn't on Facebook. But some
> people aren't on Facebook". (Participant 7)

Previous focus group studies with communities suggest that building on current provision is an appropriate starting point to improve community health and well-being [28] suggesting the importance of investing in existing community assets and not reinventing SP provision.

### 3.2. Opportunities for New Social Prescribing Interventions

During both focus groups opportunities for new SP interventions that were deliberated surrounded building upon what is already available and making current assets and provision more obvious, accessible, and approachable. In every focus group, participants implied that the Hub presented an opportunity to hold well-being events or fairs where community members could drop-in and see what is available in the community. Evidence of such events suggest their effectiveness in improving public health and health literacy [29] generating self-efficacy and confidence among community members [30] as well as providing screening services for identifying chronic illness upstream [31].

Participants suggested a need for a link worker, to take calls and enquiries independently of GP referrals and navigate residents towards available well-being and welfare provision towards the right service.

> "I think the idea of a link worker is a very powerful one, because from my
> experience, I have been receiving calls from people as a [participant's occupation]
> [ . . . ] and the fact that they're turning to me . . . there is an obvious gap there".
> (Participant 8)

This was a proposed solution to the lack of efficient advertising and hence awareness of existing groups and activities that could be contributing to the well-being community

members. Participants felt that a link worker should not only accept referrals from health professionals but from a network of social prescribers consisting of social care, third sector and voluntary officers that are interacting with individuals upstream, in their homes and communities daily. Such suggestions should be appreciated given that previous studies of SP interventions that accept referrals from additional services to GPs suggest that it is effective in empowering individuals with complex needs to independently promote their health and well-being [32]. This vision of a community link worker has been supported by studies that prove the benefit of appointing a link worker that has vast knowledge about the area's provisions to coordinate the SP. The evidence suggests that such an individual increases consumer trust in the SP intervention and consequently maintains their engagement, leading to increased well-being outcomes [33].

### 3.3. Possible Barriers to the Development of Co-Designed and Co-Produced Social Prescribing Interventions

The evidence from the current study suggests that any new intervention should overcome a set of barriers that are affecting the success of current provisions. Barriers indicated within the results include engaging volunteers to keep groups going in the long-term, which is a challenge to the sustainability of SP interventions that has been identified in previous evaluations [34]. Reference was made to the importance of evaluating interventions effectively to gain community members' buy-in and overcome the barrier of securing long-term funding. It is recognized that the need to evaluate coincides with previous findings [35,36]. Another barrier discussed during both focus groups was the lack of transport, confirming the negative effect of access to services deprivation that is suggested in the Welsh Multiple Index of Deprivation results for the Nantlle Valley [37].

> "The big issue is getting to and from places. That has a massive impact on people being able to take opportunities". (Participant 6)

However, it was suggested that the community is already tackling this barrier as many participants referred to a green transport scheme that has already been initiated in the Nantlle Valley and should be expanded.

### 3.4. Community Needs and Strategies

Many of the community's future needs and strategies for developing health and well-being outcomes were identified within the focus group findings. Within all focus groups it was implied that there is sufficient effort to protect the well-being of older generations and lack of effort to protect the well-being of working-age individuals as well as young people. The lack of provision for working adults is worrying given the evidence suggesting that this age cohort is facing increasing pressure. The evidence suggests that working-age adults' mental health and well-being is at risk due to the negative effects of increasing unemployment due to the coronavirus (COVID-19) [38–40]. Participants, who mainly consisted of working-age adults, primarily manifested a need for opportunities to socialize. This is in line with evidence indicating that having opportunities to socialize, such as in choirs, increases happiness and leads to a discovery of positive self-identity and a sense of self-improvement among working-age people [41]. The focus groups identified the need to increase the provision of support for new parents in the Nantlle Valley.

> "A lot of new parents can feel lonely, especially during this period, when they're prohibited from mixing with others. It can be a very lonely role can't it"? (Participant 12)

This finding is key given the evidence that indicates that new parents have suffered from lower self-efficacy since the pandemic [42] and that support groups have the ability to engage new parents with information as well as improve their relationship with their child [43].

The needs of young people were discussed to a greater extend within both focus groups. The communicated concerns were about their social confidence and well-being following the COVID-19 pandemic, reflecting other studies showing that children and

young people are now at increased risk of mental health issues [44] and negative effects of increased screen time [45] as a result of lockdowns. Reference was made to the ways in which some childrens' less privileged backgrounds cause them to negatively label themselves. This reflects the findings from other studies which suggested that parents' economic status or living in deprivation causes low self-esteem among young people due to their tendency to adopt their parents' low self-esteem attitudes as well as low social capital. Less privileged parents may also have less time and fewer resources to support their children than more privileged parents [46,47]. The findings from the focus groups suggest that the loss of youth clubs has led to an increase in anti-social behaviour again reflecting studies that prove that leisure boredom increases risk taking and delinquent behaviour [48].

As a result of their various concerns regarding young peoples' needs, participants therefore insinuated a need for interventions to support and increase young peoples' confidence. Such ideas included the re-establishment of a youth club and purposeful, inter-generational activities to allow young people to gain skills, and have positive experiences with other adults. However, it was emphasized during all focus groups that young people would need to be involved in the development of any intervention to be utilised by them. This was due to how participants had witnessed a sense of ownership and respect among young people towards interventions and initiatives that they had been a part of developing in the past.

> "We've done a little research with the young people and what they want is very varied, from session they like at 'Plas Silyn' [the leisure centre in Penygroes] to some of them just wanting somewhere to chill [ . . . ] there is evidence that owning a place [ . . . ] a place that they've made their own, increases their self-esteem". (Participant 8)

Such claims are supported by previous studies suggesting that co-production with young people leads to better acceptance and ownership [49]. Evidence demonstrates that such an approach can also lead to mutual respect and understanding with service providers, which increases the chances of developing positive well-being outcomes [50].

An additional issue that was suggested during all focus groups was the inclusivity of the community. During a Today Group focus group, this issue was implicated as participants referred to the self-enforced social exclusion that exists among less privileged individuals. Reference was made to a language barrier in the Nantlle Valley and instances where activities and events have been administrated through the medium of Welsh only, excluding those who were not Welsh speaking. The Future generations Groups participants explicated other groups that are at risk of being marginalised in the community due to lack of recognition such as those from the LGBTQ+ community, and minority cultural groups. However, it must be acknowledged that some participants were eager to empathize that there is an inherent strong solidarity within the community, rooted in past quarry communities.

There was no scope within the focus groups to further explore the community's dynamics and the reason why participants felt so different about the inclusivity of the community. However, such statements continue to indicate that some groups are vulnerable to the risk of loneliness and social isolation in the Nantlle Valley. As a result, there is a need for interventions to encourage the social inclusivity of such groups, as evidence suggests the possible negative effect of social isolation and loneliness [51]. The evidence particularly implies that such issues can increase morbidity and mortality due to risk of developing cardiovascular diseases and poor mental health [52]. More recent studies indicate that loneliness and social isolation can lead to increase in risk-taking behaviours [53].

The Future generations Groups also gave a sense of wider, more complicated issues that are threatening the well-being of future generations. These included many common, long-standing long-term challenges that are facing rural communities such as unemployment, lack of affordable housing [54], and poor planning legislation resulting in environmental damage. The community's long-term perspective is supported by previous studies

that indicate the negative effect of unemployment [55,56], lack of affordable housing [57], and unprotected environment [58] on individuals' well-being. Corresponding themes for ensuring the well-being of future generations with this current study included the need to maintain a healthy community spirit, increase the stock of affordable homes for local people, opportunities for every child to succeed, and a protected natural environment.

What is encouraging, however, is the desire expressed among the community to take responsibility for their own sustainable development and not rely on the local authority or public services and funding. Such strategies included raising the profile of the Nantlle Valley in order to change the attitudes of existing residents as well as attract new families to the area. Focus groups participants felt they had responsibility to come together and initiate community ventures and social enterprises to overcome any cuts in public service provision. Previous studies indicated that social enterprises can lead to many positive outcomes that could help overcome challenges, increase social connectedness, enhance confidence and self-esteem among individuals, increase employment and employability, and improve spaces and environments as well as access to services. Such outcomes suggest that social enterprises can therefore contribute to tackling social determinants of health upstream [59] and contribute to better and sustainable health and well-being outcomes for residents [60].

*3.5. Comparison between Groups*

The Today group focussed on short-termism and the challenges currently facing the community, whereas the Future generations group focussed on long-termism and the power of collective action to see past current challenges to envisage planning for a sustainable future. Prompting the 'Future generations' focus group to apply more of a long-term planning approach to their thinking on the key themes allowed the group to pursue the thoughts of extending beyond their lifetime and planning for future generations. The participants in the legacy group were more visionary and could see beyond their own immediate concerns and beyond their own generation inspired by cathedral thinking and grounded in the principles of intergenerational justice [10].

**4. Interpretation**

Applying a deliberation/citizens' assembly approach along with long-term thinking was consciously applied in this research to help shape future community development opportunities linked to the proposed Hub to identify local community needs in terms of health and well-being [24]. The Today Group and Future generations Group participants mainly discussed the Hub's potential to realise and facilitate the establishment of a holistic primary care provision through co-location of health services and well-being interventions. In addition to SP interventions, participants were eager for the Hub to offer additional health services (e.g., dental service) and new services (e.g., group therapy). Deliberations from focus groups identified that community members would benefit from easy access to traditional and preventative community health services, which could nurture resilient and health-conscious individuals projected forward. This vision is supported by studies that indicate how the co-locating of non-medical interventions, such as welfare advice [61], family-focus preventive interventions [62] and SP link workers [63], within primary care settings facilitates and increases patients utilisation of healthy behaviours in addition to generating positive well-being outcomes.

During both focus groups, it was suggested that the Hub has the potential to offer an untouched, simple centre point for not only health and well-being services and information, but also for every stratum of the community to socialize, with spaces integrated into the Hub to facilitate inter-generational activities, generating social cohesion. The Hub could therefore be seen as a much needed "third place" in the community, which is a space where individuals are at liberty from their multiple roles within society and are free to simply be their true self [64] (p. 265). Evidence suggests that protecting and developing such places has a part in encouraging health and well-being as they offer opportunities for spontaneous

social interaction, especially in deprived communities where third places (e.g., shops, youth clubs) tend to close down [65]. However, the participants promptly warned that the Hub should not take away from other community venues or third places, and cause everything to become too centralized. This caution is supported by previous evidence suggesting that centralizing services excessively can hinder their accessibility to those living on the outskirts of the area, posing a risk to their health and well-being [28].

## 5. Study Limitations

This study's findings are limited as the sample was small, and not fully representative of the Nantlle Valley community in terms of age, gender, nor race. However, there was extensive discussion regarding issues affecting young people and some references to provision for older people during the focus groups, as well as groups that participants considered to be excluded from the community. As a result, although these groups were underrepresented, the data continues to provide insights into issues affecting them and strategies that could lead to positive well-being outcomes among them.

It must be considered that the COVID-19 pandemic has posed limitations on this study. An additional factor affecting the representativeness of the data is the fact that Nantlle Valley residents that did not have home access to the internet, or limited internet access at home were excluded since the study had to be conducted remotely. In addition, the severity of the pandemic during February 2021 meant that the external circumstances were uncertain and as a result arguably impacted on participants' ability to look ahead and think about future strategies.

In terms of data collection methods, although the researchers chose focus groups as the most suitable method for producing naturalistic data on the community's collective attitudes and perspectives, it is recognized that any group precisely gathered or facilitated is not a completely naturalistic setting and participants were aware throughout the discussion that their contribution was being treated as data [66]. Although the researchers utilized moderating methods that encouraged natural conversation and a relaxed atmosphere that appreciated each participant's contribution, it is recognized that there is still a risk that participants might have modified their answers to be sociably accepted responses, affecting the reliability of the results [67].

The cohort age range may also be a limitation on taking a co-production approach to intervention development based on individual and collective needs across generations. Balancing needs and expectations can be challenging when taking a co-designed, co-produced approach when developing SP interventions for long term sustainability, given the cognitive leap to long-term thinking [10]. This is based on the requirement that social co-operation involves imaginative capacity to see into the future.

With regards to reciprocity, it is acknowledged that this cohort of participants will not be involved in the development of future opportunities, however, their influence in thinking towards the future is shaping the current SP interventions associated with the Hub, with long-term thinking for health and well-being for future generations.

## 6. Conclusions

The primary data collection of this study initiated a conversation amongst the residents of the Nantlle Valley around SP interventions and the potential of a new health and well-being Hub to deliver positive health and well-being outcomes in the community. There was limited knowledge about current SP interventions within the Nantlle Valley and participants expressed regret at lack of awareness.

Findings from all the focus groups indicated that the Hub has the potential, not only to improve the health and well-being outcomes of the community through an innovative, holistic provision, but also to improve community well-being by providing much needed opportunities to encourage social interaction between different strata of the community. The results therefore suggest that opportunities to socialise are considered equally important to the well-being of residents as any medical or non-medical intervention. In line with the

Hub's vision to regenerate the area and contribute to realizing the goals set out within the Well-Being of Future Generations Act (Wales) 2015 [5]; findings from this study indicate that taking a future generations thinking approach to SP interventions such as the Hub, can build healthier, cohesive, inclusive communities generating a vibrant culture and leading to resilient and prosperous communities.

The focus groups method in this study was shaped by an overarching objective of determining if and how a long-term thinking approach to deliberations with communities produces data that identifies opportunities for sustainable interventions that can benefit the well-being of future generations. Drawing upon the principles of Future Design research methods and citizen assemblies in developing sustainable strategies, this study consisted of deliberations with the community of the Nantlle Valley. Deriving from the benefits of Future Design approaches in encouraging participants to think long-term, a novel approach was applied to the focus groups. The "Today Group" deliberated on the well-being of the community today, and the "Future generations Group" deliberated on the well-being of future generations facilitated by the future-ahead-and-back mechanism [21]. The results of this research suggest that such a deliberating method was appropriate but requires strengthening through future research. This is mainly due to the tendency among participants to return to short-term thinking, circling back to issues that affected the community 'today' when COVID-19 restrictions were still in place. To take account of this finding, future research applying this approach should include a workshop at the beginning of the focus groups that sets the future generations frame of mind. Another option would be to draw upon recent Future Design studies, hold workshops face to face, and divide participants into two groups with one group representing future generations throughout the discussion. Evidence suggests that the presence of such an imaginary future generation leads participants to make the most sustainable choices that benefit future generations [68,69].

Barriers to the development of co-designed and co-produced SP interventions include lack of buy-in into interventions and lack of volunteers to ensure that any intervention is sustainable. There is also a need to evaluate interventions, however this is a barrier because SP interventions are linked with short-term funding making longitudinal assessments difficult to conduct. The results from this study demonstrate a scope for new interventions in the Nantlle Valley to make current interventions more accessible and tackle challenges such as lack of inclusivity in the community, and the different well-being needs of different age groups.

Applying long-term thinking to citizens' assembly deliberation processes delivers a voice to the interests of future generations, providing inter-generational equity.

Well-being of future generations is gaining momentum as demonstrated by the Well-being of Future Generations (Wales) Act 2015 [5] which scrutinises the effects of decisions on future generations to ensure the impacts of decisions are fully considered [70]. The act is revolutionary, as it places a legal obligation to consider the well-being of current and future people when making policy decisions. The Act dictates five ways of working to reach decisions, including prevention, long-termism, collaboration, participation, and integrating activities. Citizens' assemblies approaches used in modern democracies such as Wales and Ireland have guardians and stewards who ensure that the voices and interests of future generations are put into action [71]. Citizens' assemblies can be extremely effective at transcending short-term thinking, demonstrating equity, equality, and fairness, and are reflective of long-term issues facing society [10].

This study provides insights into participatory deliberative methods, comprising citizens' assemblies which take long-term thinking into account. Deliberation findings from this local community study indicate that the application of The Good Ancestor Principles [10] facilitates Future Design methods, drives concepts such as cathedral thinking [72], and leads to insights that can shape sustainable, healthy communities for future generations.

**Author Contributions:** Conceptualization, M.L., L.H.S. and G.M.T.; methodology, M.L., L.H.S. and G.M.T.; resources, G.M.T., M.L. and L.H.S.; formal analysis, G.M.T.; writing—original draft preparation, review and editing, M.L., L.H.S. and R.T.E.; visualization, G.M.T.; supervision, M.L. and L.H.S. All authors have read and agreed to the published version of the manuscript.

**Funding:** G.M.T. was a Master of Research (MRes) student funded by The European Social Fund through a Knowledge Economy Skills Scholarship (KESS2 East 80815).

**Institutional Review Board Statement:** Ethical approval for this study was granted by Bangor University's Healthcare and Medical Sciences Academic Ethics Committee (2020–16850) on 11 January 2021. Due to the COVID-19 pandemic restrictions, the participants were sent an electronic consent form to return before the focus group and their verbal consent were recorded at the beginning of each focus group.

**Data Availability Statement:** The data presented in this study are available on request from the corresponding author.

**Acknowledgments:** The authors wish to thank Andrew Rogers at Gwenallt Consulting and Mair Edwards at Grŵp Cynefin Housing Association for their valuable comments throughout the production of this study. The authors are grateful to the Welsh School of Social Prescribing Research for the opportunity to present the findings of an early draft and for their constructive comments. The authors are also very grateful to Knowledge Economy Skills Scholarships (KESS2 East) for funding the MRes.

**Conflicts of Interest:** The authors declare that the research was conducted in the absence of any commercial or financial relationships that could be construed as a potential conflict of interest.

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
