# Peer review of "Intergenerational Deliberations for Long Term Sustainability"

_challenges, doi:10.3390/challe14010011_

Round 1

Reviewer 1 Report

This paper covers some very interesting ground, engaging with important issues around community input into policy/practice design, holistic approaches to health and wellbeing, and how people think about future generations. However, the issues are not linked and drawn together in a coherent way and the paper would only be suitable for publication with major revisions.

The fundamental problem with the paper is that the link between the Hub and the wellbeing of future generations is not established. This leaves the reader questioning the whole design of the study - if the Hub's actual or potential relevance to the wellbeing of future generations is not clear, why ask people to reflect on it?

The future thinking frameworks used by the researchers in the future-oriented focus groups are largely intended to promote consideration of impacts on people some way into the future, e.g. 50 years or more. What impact can the Hub have on the wellbeing of people 50 years in the future? Is it trying to have an impact? Through what activities? There is possibly a case to be made (e.g. reducing intergenerational transmission of health problems, targeted strategies to improve the long-term health of very young children) but it is not made.

Even if this link can be established, it was presumably not made clear to the future focus group participants, therefore they could not explicitly engage with how the Hub can support the wellbeing of future people, or whether it should be attempting to do so. And the really interesting issue - whether resources should be redirected from present to future needs (Kzarnic's intergenerational justice principle - and there is an extensive literature on this issue in disciplines as wide-ranging as political theory, philosophy, environmental science and accounting) - is not canvassed at all.

A couple of times the paper comes close to directly addressing the issue of what we can/should do now for future people (e.g. at lines 433-35) but then stops short of saying anything meaningful about what the focus group data tells us in this regard.

Suggested reframing

Depending on the nature of the focus group data, it may be better to reframe the paper as simply a discussion of community members' views on the Hub, based on citizens' assembly style co-production. This could include what they would like it to deliver not just now but in the short to medium term, i.e. sustainability with a 5-15 year horizon rather than longer. There is possibly also scope for some reflection on the value of citizens' assembly style co-production of a policy/practice initiative like the Hub.

Comments on methods

- Clarify how citizens' assemblies are different to focus groups.

- Clarify how the sampling was 'purposeful' - it didn't appear to be.

- The theoretical and methodological framing seems a bit 'scattershot' - how are concepts like citizens' assemblies, FAB, the Good Ancestor framework, etc linked? It's not clear how the first two were actually deployed in the research.

- Move 2.4 up to sit with the discussion of participants and sampling.

- Line 351 - the reason the sample is not representative of the broader population is not because it failed to include different ethnic groups but because it was a very small sample and could not possibly be representative.

Comments on findings and discussion

- State a clear research aim or question early on (in the introduction) and make sure it aligns with the methods, analysis and discussion of results. The two different focus groups were asked two separate sets of questions, suggesting there were two different research questions being addressed, though it is not completely clear what either of them were; nor was it clear how the four themes were identified.

- Include some quotes from focus group discussion to illustrate key points.

- The comment at lines 297-99 suggests the present study doesn't add much that is new or interesting.

- Avoid claiming findings that are not adequately justified by the results described (e.g. lines 324-27, lines 430-32, lines 435-40).

Minor points

- The manuscript could do with a proof read and some restructuring to improve flow and make it easier to follow.

- Some concepts are poorly defined (e.g. social prescribing) or not defined at all (the Hub, Grŵp Cynefin).  

Author Response

Dear Reviewer 1

Many thanks for your knowledgeable and detailed comments on the first draft of our Intergenerational deliberation for long term sustainability paper for the Challenges journal.

We have incorporated your feedback into the revised draft and sincerely hope that we have responded to your concerns regarding the missing details. We have amened every section including Abstract, Introduction, Materials, Findings, Interpretaton and Conclusions. We have also made changes to the Reference section. Track changes were used throughout the document.

We anticipate that the changes will meet your required standards for publication. Thanking you in advance.

Kind regards

Dr Llinos Haf Spencer and Professor Mary Lynch

On behalf of the research team.

Author Response

Dear Reviewer 2

Many thanks for your comprehensive review of our paper. This research study was completed in summer 2021 and therefore it is not possible to collect any more data or indeed conduct statistical analyses on the qualitative data that was collected.

We have improved the readability of the paper by changing the abstract, methods, and findings sections to make it easier for a general audience to understand the methodology utilized. We have also made changes to the Findings and Interpretation sections in line with your and Reviewer 1's comments.

We appreciate your detailed and knowledgeable feedback, which has helped us to shape the revised manuscript.

Kind regards

Dr Llinos Haf Spencer and Professor Mary Lynch

On behalf of the wider team

Round 2

Reviewer 1 Report

Thank you for incorporating revisions to this paper. I believe the changes made have improved the quality of the submission.

The revisions give a better understanding of key concepts like social prescribing and the Hub. The links made between the Hub (including its intergenerational activities) and the Wellbeing of Future Generations Act strengthen the case for exploring the application of long-term thinking in this context.

A key contribution of the article - exploring community members’ willingness to consider long-term/future interests in health and wellbeing policy/programming - has been made clearer. This contribution is both interesting and important, not least because public preferences are often a barrier to governments taking the long view in policymaking.

The additions at lines 78-80 and 104-106 help to explain why and how the citizens’ assembly and Future Design approaches were used.

Lines 111-118 clearly state aims and objectives, which is an important improvement.

Table 2 provides helpful clarification of the difference between the two focus groups.

The addition of participant quotes in the results section is welcome and provides some nice illustrations of the points emerging from the thematic analysis.

The addition at lines 369 to 376 is helpful but would fit better earlier on, probably in the methods section, rather than in the interpretation of results section.

Remaining issues

Methods: I still find it difficult to see how the focus groups were citizens’ assemblies and I think this needs to be clarified.

More fundamentally, the usefulness of applying a long-term thinking approach in some focus groups but not others does not emerge from the discussion of results, calling into question the framing of the paper in terms of Future Design and principles of intergenerational sustainability. The discussion needs to articulate what differences emerged between the two sets of focus groups and what the significance of these differences was. If there were no real differences, the paper needs to address the significance of this, e.g. asking questions to prompt long-term thinking didn’t make much difference to the focus group discussion so should we then be considering other ways of prompting such thinking?

The paper needs a close proof read as there are quite a few minor typographical errors. For example, just on the first page:

Line 24, delete one of the full stops.

Check for consistent use of citizen assembly, citizens assembly, citizen’s assembly or citizens’ assembly throughout – the latter is probably best.

Line 31, delete one of the spaces.

Line 33, self-manages should be self-management.

Line 38, replace accords with accord.

Line 39, replace Governments with Government’s.

Line 42, replace and with an.

Reviewer 2 Report

Please address the previous comments 4-7. I do not see them reflected in the revised manuscript. Literature review, to support what is new, and where the study contributes to the existing literature.  I strongly recommend reconsidering it. 
